# *AdaWAC*: Adaptively Weighted Augmentation Consistency Regularization for Volumetric Medical Image Segmentation

## Abstract

Sample reweighting is an effective strategy for learning from training data coming from a mixture of different subpopulations. However, existing reweighting algorithms do not fully take advantage of the particular type of data distribution encountered in volumetric medical image segmentation, where the training data images are uniformly distributed but their associated data labels fall into two subpopulations—"label-sparse" and "label-dense"—depending on whether the data image occurs near the beginning/end of the volumetric scan or the middle. For this setting, we propose *AdaWAC* as an adaptive weighting algorithm that assigns label-dense samples to supervised cross-entropy loss and label-sparse samples to unsupervised consistency regularization. We provide a convergence guarantee for *AdaWAC* by appealing to the theory of online mirror descent on saddle point problems. Moreover, we empirically demonstrate that *AdaWAC* not only enhances segmentation performance and sample efficiency but also improves robustness to the subpopulation shift in labels.

## 1 Introduction

Modern machine learning has been revolutionizing the field of medical imaging, especially in computer-aided diagnosis with Computed Tomography (CT) and Magnetic Resonance Imaging (MRI) scans. While the successes of most classical learning algorithms (*e.g.*, empirical risk minimization (ERM)) build upon the assumption that training samples are independently and identically distributed (*i.i.d.*), real-world volumetric medical images rarely fit into this picture. Specifically for medical image segmentation, as instantiated in Figure 1, the segmentation labels corresponding to different cross-sections of organs within a given volume tend to have distinct distributions. That is, the slices toward the beginning/end of the volume that contain no target organs have very few, if any, segmentation labels (which we refer to as "label-sparse"); whereas segmentation labels are prolific in the slices toward the middle of the volume ("label-dense"). Such discrepancy in labels results in distinct difficulty levels measured by the training cross-entropy (Wang et al., 2021b) and leads to various training schedulers (Tullis & Benjamin, 2011; Tang et al., 2018; Hacohen & Weinshall, 2019). Motivated by the separation between label-sparse and label-dense samples, we explore the following questions in this work:

> *What is the effect of separation between sparse and dense labels on segmentation?*
> *Can we leverage such separation to improve the segmentation accuracy?*

We first formulate the mixture of label-sparse and label-dense samples as a subpopulation shift in the conditional distribution of labels given images $P(\mathbf{y}|\mathbf{x})$. As illustrated in Figure 1, such subpopulation shift induces a separation in supervised cross-entropy between sparse and dense labels despite the uniformity of data images.

Prior works address the subpopulation shift issue by utilizing hard thresholding algorithms such as Trimmed Loss Estimator (Shen & Sanghavi, 2019), MKL-SGD (Shah et al., 2020), Ordered SGD (Kawaguchi & Lu, 2020), and quantile-based Kacmarz algorithm (Haddock et al., 2020). However, these trimmed-loss-based methods discard the samples from some subpopulations (*e.g.* samples with label corruption estimated by their losses) at each iteration, which results in loss of

information in the discarded data, leading to reduced sample efficiency. Relaxing the hard thresholding operator to soft thresholding is proposed to incorporate the information from both subpopulations (Wang et al., 2018; Sagawa et al., 2020). However, lowering the weights assigned to some subpopulations of data according to the properties of their labels reduces the importance of the data and labels simultaneously, suggesting that we may further improve the learning efficiency by exploiting the uniformity of data and the separation of labels separately.

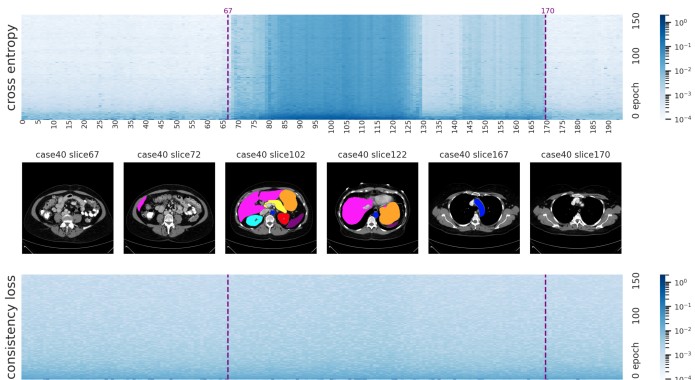

Figure 1: Evolution of cross-entropy losses versus consistency regularization terms for slices across one training volume (Case 40) in the Synapse dataset (Section 5) during training.

Instead of thresholding out or down-weighting the label-sparse samples, we exploit the image inputs of these samples via *augmentation consistency regularization*. Consistency regularization (Bachman et al., 2014; Laine & Aila, 2016; Sohn et al., 2020) aims to learn the proximity between data augmentations of the same samples; crucially, this set-up does not involve the data labels, and hence consistency regularization has become an essential strategy for utilizing unlabeled data. For medical imaging tasks, consistency regularization has been extensively studied in the semi-supervised learning setting (Bortsova et al., 2019; Zhao et al., 2019; Li et al., 2020; Wang et al., 2021a; Zhang et al., 2021; Zhou et al., 2021; Basak et al., 2022). By contrast, we will explore its potency in the fully supervised setting—leveraging the spare information in all image inputs, regardless of their label subpopulations.

Moreover, in light of the uniformity of unsupervised consistency on different slices throughout each volume, the augmentation consistency of the encoder layer outputs serves as a natural reference for separating samples from different subpopulations. Whereby, we introduce the *weighted augmentation consistency (WAC) regularization*—a minimax formulation that not only incorporates the consistency regularization but also leverages the consistency regularization as a reference for reweighting the cross-entropy and the augmentation consistency terms corresponding to different samples. At the saddle point, the WAC regularization automatically separates samples from different label subpopulations by assigning all weight to the consistency terms for label-sparse samples, and all weight to the cross-entropy terms for label-dense samples.

Furthermore, as an algorithm for solving the minimax problem posed by the WAC regularization, we propose *AdaWAC*—an *adaptive weighting* scheme inspired by a mirror-descent-based algorithm for distributionally-robust optimization (Sagawa et al., 2020). By adaptively adjusting the weights between the cross-entropy and consistency terms of different samples, *AdaWAC* comes with both a convergence guarantee and empirical success.

Overall, we summarize the main contributions of this work as follows:

- We cast the discrepancy between the sparse and dense labels within each volume as a subpopulation shift in the conditional distribution $P(\mathbf{y}|\mathbf{x})$ (Section 2).
- We propose WAC regularization which uses the consistency of encoder layer outputs (in a UNet architecture) as a natural reference to incentivize separation between samples with sparse and dense labels (Section 3), along with an adaptive weighting algorithm—*AdaWAC*—for solving the WAC regularization problem with a convergence guarantee (Section 4).
- We empirically demonstrate the potency of *AdaWAC* not only in enhancing segmentation performance and sample efficiency but also in improving distributional robustness (Section 5).

## 1.1 RELATED WORK

**Sample reweighting.** Sample reweighting is a popular strategy for coping with subpopulation shifts in training data where different weights are assigned to samples from different subpopulations. In particular, the distributionally-robust optimization (DRO) (Ben-Tal et al., 2013; Duchi et al., 2016; Duchi & Namkoong, 2018; Sagawa et al., 2020) considers a collection of training sample groups from different distributions where, with the explicit grouping of samples, the goal is to minimize the worst-case loss over the groups. Without prior knowledge on sample grouping, importance sampling (Needell et al., 2014; Zhao & Zhang, 2015; Alain et al., 2015; Loshchilov & Hutter, 2015; Gopal, 2016; Katharopoulos & Fleuret, 2018), iterative trimming (Kawaguchi & Lu, 2020; Shen & Sanghavi, 2019), and empirical-loss-based reweighting (Wu et al., 2022) are commonly incorporated in the stochastic optimization process for adaptive reweighting and separation of samples from different subpopulations.

**Consistency regularization.** Consistency regularization (Bachman et al., 2014; Laine & Aila, 2016; Sohn et al., 2020; Berthelot et al., 2019) is a popular way to exploit data augmentations that encourage the model to learn the vicinity among augmentations of the same sample, with the assumption that data augmentations generally preserve the semantic information in data.

For medical imaging, consistency regularization is generally leveraged as a semi-supervised learning tool (Bortsova et al., 2019; Zhao et al., 2019; Li et al., 2020; Wang et al., 2021a; Zhang et al., 2021; Zhou et al., 2021; Basak et al., 2022). In efforts to incorporate consistency regularization in medical image segmentation with augmentation-sensitive labels, Li et al. (2020) encourages transformation consistency between predictions with augmentations applied to the image inputs and the segmentation outputs. Basak et al. (2022) penalizes inconsistent segmentation outputs between teacher-student models, with MixUp (Zhang et al., 2017) applied on image inputs of the teacher model and segmentation outputs of the student model. Instead of enforcing consistency in the segmentation output space as above, our algorithm leverages the insensitivity of sparse labels to augmentations and encourages consistent encodings (in the latent space of encoder outputs) on label-sparse samples.

## 2 PROBLEM SETUP

**Notations.** We denote $[K] = \{1, \ldots, K\}$ for any $K \in \mathbb{N}$. For an arbitrary tensor, we adapt the syntax for Python slicing on the subscript (except counting from 1) to represent its elements and subtensors. For example, $\mathbf{x}_{[i,j]}$ denotes the $(i,j)$-entry of the two-dimensional tensor $\mathbf{x}$, and $\mathbf{x}_{[i,:]}$ denotes its $i$-th row. Let $\mathbb{I}$ be a function onto $\{0,1\}$ such that, for any event $e$, $\mathbb{I}\{e\} = 1$ if $e$ is true and 0 otherwise. For any distribution $P$ and $n \in \mathbb{N}$, let $P^n$ denote the joint distribution of $n$ samples drawn *i.i.d.* from $P$. We refer to an event as happening with high probability (*w.h.p.*) if it takes place with probability $1 - \Omega\left(\text{poly}\left(n\right)\right)^{-1}$.

### 2.1 PIXEL-WISE CLASSIFICATION WITH SPARSE AND DENSE LABELS

We consider the volumetric medical image segmentation as a pixel-wise multi-class classification problem where we aim to learn a pixel-wise classifier $h : \mathcal{X} \to [K]^d$ that serves as a good approximation for the ground truth $h^* : \mathcal{X} \to [K]^d$.

Recall the separation of cross-entropy losses between samples with different fractions of non-background labels during training from Figure 1. We refer to a sample $(\mathbf{x}, \mathbf{y}) \in \mathcal{X} \times [K]^d$ as *label-sparse* if most pixels in $\mathbf{y}$ are labeled as the background such that the cross-entropy loss on $(\mathbf{x}, \mathbf{y})$ converges rapidly in the early stage of training[1]. Otherwise, we say that $(\mathbf{x}, \mathbf{y})$ is *label-dense*. Formally, we describe such variation as a subpopulation shift in the label distribution.

**Definition 1** (Mixture of label-sparse and label-dense distributions)**.** Let $P_0$ and $P_1$ be the distributions of *label-sparse and label-dense samples* with distinct conditional distributions $P_0\left(\mathbf{y}|\mathbf{x}\right)$

---

[1]Although the sparsity of non-background pixels in the segmentation label is a key feature of label-sparse samples (as the name suggests), the unknown cut-off on such sparsity degenerates it as a sufficient condition for the rapid convergence of cross-entropy loss (Figure 1). Instead of making distinction with the sparsity of non-background pixels, we formalize a natural separation between label-sparse and label-dense samples in Assumption 1, based on which our algorithm can distinguish different samples spontaneously.

and $P_1(\mathbf{y}|\mathbf{x})$, respectively, but the same marginal distribution $P(\mathbf{x})$ such that $P_i(\mathbf{x}, \mathbf{y}) = P_i(\mathbf{y}|\mathbf{x}) P(\mathbf{x})$ $(i = 0, 1)$. For $\xi \in [0, 1]$, we define a data distribution $P_\xi$ where each sample $(\mathbf{x}, \mathbf{y}) \sim P_\xi$ is drawn either from $P_1$ with probability $\xi$ or from $P_0$ with probability $1 - \xi$.

We aim to learn a pixel-wise classifier from a function class $\mathcal{H} \ni h_\theta = \mathrm{argmax}_{k \in [K]} f_\theta(\mathbf{x})_{[j,:]}$ for all $j \in [d]$ where the underlying function $f_\theta \in \mathcal{F}$, parameterized by some $\theta \in \mathcal{F}_\theta$, admits an encoder-decoder structure:

$$\mathcal{F} = \left\{ f_\theta = \phi_\theta \circ \psi_\theta \mid \phi_\theta : \mathcal{X} \to \mathcal{Z}, \ \psi_\theta : \mathcal{Z} \to [0, 1]^{d \times K} \right\}. \tag{1}$$

Here $\phi_\theta, \psi_\theta$ correspond to the encoder and decoder functions, respectively; $(\mathcal{F}_\theta, \langle \cdot, \cdot \rangle_\mathcal{F})$ denotes the inner product space of parameters with the induced norm $\|\cdot\|_\mathcal{F}$ and dual norm $\|\cdot\|_{\mathcal{F},*}$; $(\mathcal{Z}, \varrho)$ is a latent metric space.

To learn from segmentation labels, we consider the *averaged cross-entropy loss*:

$$\ell_{CE}(\theta; (\mathbf{x}, \mathbf{y})) = -\frac{1}{d} \sum_{j=1}^{d} \sum_{k=1}^{K} \mathbb{I}\{\mathbf{y}_{[j]} = k\} \cdot \log\left(f_\theta(\mathbf{x})_{[j,k]}\right) = -\frac{1}{d} \sum_{j=1}^{d} \log\left(f_\theta(\mathbf{x})_{[j,\mathbf{y}_{[j]}]}\right). \tag{2}$$

We assume proper learning where there exists $\theta^* \in \bigcap_{\xi \in [0,1]} \mathrm{argmin}_{\theta \in \mathcal{F}_\theta} \mathbb{E}_{(\mathbf{x}, \mathbf{y}) \sim P_\xi}[\ell_{CE}(\theta; (\mathbf{x}, \mathbf{y}))]$ that is invariant with respect to $\xi$.

## 2.2 AUGMENTATION CONSISTENCY REGULARIZATION

Despite the invariance of $f_{\theta^*}$ to $P_\xi$ on the population loss, in practice we have a finite number of training samples and the predominance of label-sparse samples in the training set introduces difficulties due to the class imbalances. As a specific extreme scenario for the pixel-wise classifier with encoder-decoder structure (Equation (1)), when the label-sparse samples are predominant ($\xi \ll 1$), a decoder function $\psi_\theta$ that predicts every pixel as background can achieve near-optimal cross-entropy loss, regardless of what the encoder function $\phi_\theta$ is, considerably compromising the test performance (*cf.* Table 1). To encourage legit encoding even in absence of sufficient dense labels, we leverage the unsupervised consistency regularization on the *encoder function* $\phi_\theta$ based on data augmentations.

Let $\mathcal{A}$ be a distribution over transformations on $\mathcal{X}$ where for any $\mathbf{x} \in \mathcal{X}$, each $A \sim \mathcal{A}$ ($A : \mathcal{X} \to \mathcal{X}$) induces an augmentation $A(\mathbf{x})$ of $\mathbf{x}$ that perturbs low-level information in $\mathbf{x}$. We aim to learn an encoder function $\phi_\theta : \mathcal{X} \to \mathcal{Z}$ that is capable of filtering out low-level information from $\mathbf{x}$ and therefore provides similar encodings for augmentations of the same sample. Recalling the metric $\varrho$ on $\mathcal{Z}$, for a given scaling hyperparameter $\lambda_{AC} > 0$, we measure the similarity between augmentations with a consistency regularization term on $\phi_\theta(\cdot)$: for any $A_1, A_2 \sim \mathcal{A}^2$,

$$\ell_{AC}(\theta; \mathbf{x}, A_1, A_2) \triangleq \lambda_{AC} \cdot \varrho\Big(\phi_\theta(A_1(\mathbf{x})), \phi_\theta(A_2(\mathbf{x}))\Big). \tag{3}$$

For the $n$ training samples $\{(\mathbf{x}_i, \mathbf{y}_i)\}_{i \in [n]} \sim P_\xi^n$, we consider $n$ pairs of data augmentation transformations $\{(A_{i,1}, A_{i,2})\}_{i \in [n]} \sim \mathcal{A}^{2n}$. In the basic version, we encourage the similar encoding $\phi_\theta(\cdot)$ of the augmentation pairs $(A_{i,1}(\mathbf{x}_i), A_{i,2}(\mathbf{x}_i))$ for all $i \in [n]$ via consistency regularization:

$$\min_{\theta \in \mathcal{F}_{\theta^*}(\gamma)} \frac{1}{n} \sum_{i=1}^{n} \ell_{CE}(\theta; (\mathbf{x}_i, \mathbf{y}_i)) + \ell_{AC}(\theta; \mathbf{x}_i, A_{i,1}, A_{i,2}). \tag{4}$$

We enforce consistency on $\phi_\theta(\cdot)$ in light of the encoder-decoder architecture: the encoder is generally designed to abstract essential information and filters out low-level non-semantic perturbations (*e.g.*, those introduced by augmentations), while the decoder recovers the low-level information for the pixel-wise classification. Therefore, with different $A_1, A_2 \sim \mathcal{A}$, the encoder output $\phi_\theta(\cdot)$ tends to be more consistent than the other intermediate layers, especially for label-dense samples.

## 3 WEIGHTED AUGMENTATION CONSISTENCY (WAC) REGULARIZATION

As the motivation, we begin with a key observation about the averaged cross-entropy:

*Remark* 1 (Separation of averaged cross-entropy loss on $P_0$ and $P_1$). As demonstrated in Figure 1, the sparse labels from $P_0$ tend to be much easier to learn than the dense ones from $P_1$, leading to considerable separation of averaged cross-entropy losses on the sparse and dense labels after a sufficient number of training epochs – $\ell_{CE}(\theta; (\mathbf{x}, \mathbf{y})) \ll \ell_{CE}(\theta; (\mathbf{x}', \mathbf{y}'))$ for most label-sparse samples $(\mathbf{x}, \mathbf{y}) \sim P_0$ and label-dense samples $(\mathbf{x}', \mathbf{y}') \sim P_1$.

Although Equation (4) with consistency regularization alone can boost the segmentation accuracy during testing (*cf.* Table 4), it does not take the separation between label-sparse and label-dense samples into account. In Section 5, we will empirically demonstrate that proper exploitation of such separation, like the formulation introduced below, can bring compelling further improvements.

Concretely, we formalized the notion of separation between $P_0$ and $P_1$ based on the consistency regularization term (Equation (3)) with the following assumption [2].

**Assumption 1** ($N$-separation between $P_0$ and $P_1$). Given a sufficiently small $\gamma > 0$, let $\mathcal{F}_{\theta^*}(\gamma) = \{\theta \in \mathcal{F}_\theta \mid \|\theta - \theta^*\|_{\mathcal{F}} \leq \gamma\}$ be a compact and convex neighborhood of well-trained pixel-wise classifiers[3]. We say that $P_0$ *and* $P_1$ *are $N$-separated over* $\mathcal{F}_{\theta^*}(\gamma)$ if there exists $\omega > 0$ such that both:

(i) $\ell_{CE}(\theta; (\mathbf{x}, \mathbf{y})) < \ell_{AC}(\theta; \mathbf{x}, A_1, A_2)$ for all $\theta \in \mathcal{F}_{\theta^*}(\gamma)$ given $(\mathbf{x}, \mathbf{y}) \sim P_0$
(ii) $\ell_{CE}(\theta; (\mathbf{x}, \mathbf{y})) > \ell_{AC}(\theta; \mathbf{x}, A_1, A_2)$ for all $\theta \in \mathcal{F}_{\theta^*}(\gamma)$ given $(\mathbf{x}, \mathbf{y}) \sim P_1$

hold with probability $1 - \Omega\left(N^{1+\omega}\right)^{-1}$ over $((\mathbf{x}, \mathbf{y}), (A_1, A_2)) \sim P_\xi \times \mathcal{A}^2$.

This assumption is motivated by the empirical observation that the perturbation in $\phi_\theta(\cdot)$ induced by $\mathcal{A}$ is more uniform across $P_0$ and $P_1$ than the averaged cross-entropy, as instantiated in Figure 3.

Under Assumption 1, up to a proper scaling hyperparameter $\lambda_{AC}$, the consistency regularization (Equation (3)) can separate the averaged cross-entropy loss (Equation (2)) on $N$ label-sparse and label-dense samples with probability $1 - \Omega\left(N^\omega\right)^{-1}$ by the union bound (as explained formally in Appendix A). In particular, the larger $N$ correspond to the stronger separation between $P_0$ and $P_1$.

With Assumption 1, we introduce a minimax formulation that incentivizes the separation of label-sparse and label-dense samples automatically by introducing a flexible weight $\boldsymbol{\beta}_{[i]} \in [0, 1]$ that balances $\ell_{CE}(\theta; (\mathbf{x}_i, \mathbf{y}_i))$ and $\ell_{AC}(\theta; \mathbf{x}_i, A_{i,1}, A_{i,2})$ for each of the $n$ samples.

$$\widehat{\theta}^{WAC}, \widehat{\boldsymbol{\beta}} \in \underset{\theta \in \mathcal{F}_{\theta^*}(\gamma)}{\operatorname{argmin}} \underset{\boldsymbol{\beta} \in [0,1]^n}{\operatorname{argmax}} \left\{\widehat{L}^{WAC}(\theta, \boldsymbol{\beta}) \triangleq \frac{1}{n} \sum_{i=1}^n \widehat{L}_i^{WAC}(\theta, \boldsymbol{\beta})\right\} \tag{5}$$

$$\widehat{L}_i^{WAC}(\theta, \boldsymbol{\beta}) \triangleq \boldsymbol{\beta}_{[i]} \cdot \ell_{CE}(\theta; (\mathbf{x}_i, \mathbf{y}_i)) + (1 - \boldsymbol{\beta}_{[i]}) \cdot \ell_{AC}(\theta; \mathbf{x}_i, A_{i,1}, A_{i,2}).$$

With convex and continuous loss and regularization terms (formally in Proposition 1), Equation (5) has a saddle point where $\widehat{\boldsymbol{\beta}}$ separates the label-sparse and label-dense samples under Assumption 1.

**Proposition 1** (Formal proof in Appendix A). *Assume that* $\ell_{CE}(\theta; (\mathbf{x}, \mathbf{y}))$ *and* $\ell_{AC}(\theta; \mathbf{x}, A_1, A_2)$ *are convex and continuous in* $\theta$ *for all* $(\mathbf{x}, \mathbf{y}) \in \mathcal{X} \times [K]^d$ *and* $A_1, A_2 \sim \mathcal{A}^2$; $\mathcal{F}_{\theta^*}(\gamma) \subset \mathcal{F}_\theta$ *is compact and convex. If* $P_0$ *and* $P_1$ *are $n$-separated (Assumption 1), then there exists* $\widehat{\boldsymbol{\beta}} \in \{0,1\}^n$ *and* $\widehat{\theta}^{WAC} \in \operatorname{argmin}_{\theta \in \mathcal{F}_{\theta^*}(\gamma)} \widehat{L}^{WAC}(\theta, \widehat{\boldsymbol{\beta}})$ *such that*

$$\min_{\theta \in \mathcal{F}_{\theta^*}(\gamma)} \widehat{L}^{WAC}(\theta, \widehat{\boldsymbol{\beta}}) = \widehat{L}^{WAC}(\widehat{\theta}^{WAC}, \widehat{\boldsymbol{\beta}}) = \max_{\boldsymbol{\beta} \in [0,1]^n} \widehat{L}^{WAC}(\widehat{\theta}^{WAC}, \boldsymbol{\beta}). \tag{6}$$

*Further,* $\widehat{\boldsymbol{\beta}}$ *separates the label-sparse and label-dense samples—*$\boldsymbol{\beta}_{[i]} = \mathbb{I}\{(\mathbf{x}_i, \mathbf{y}_i) \sim P_1\}$*—w.h.p..*

In other words, for $n$ samples drawn from a mixture of the $n$-separated $P_0$ and $P_1$, at the saddle point, Equation (5) automatically identifies the label-sparse samples with $\boldsymbol{\beta}_{[i]} = 0$, learning more from the unsupervised consistency regularization, and the label-dense ones with $\boldsymbol{\beta}_{[i]} = 1$, emphasizing more on the supervised averaged cross-entropy loss.

---

[2]We note that although Assumption 1 can be rather strong, it is only required for the separation guarantee of label-sparse and label-dense samples with high probability in Proposition 1, but not for the adaptive weighting algorithm introduced in Section 4 or in practice for the experiments.

[3]With pretrained initialization, we assume that the optimization algorithm is always probing in $\mathcal{F}_{\theta^*}(\gamma)$.

## 4  ADAPTIVELY WEIGHTED AUGMENTATION CONSISTENCY (*AdaWAC*)

*Remark* 2 (Connection to hard thresholding algorithms). The saddle point of Equation (5) is closely related to hard thresholding algorithms like Ordered SGD (Kawaguchi & Lu, 2020) and iterative trimmed loss (Shen & Sanghavi, 2019). In each iteration, these algorithms update the model only on a proper subset of training samples based on the (ranking of) current empirical risks. Compared to hard thresholding algorithms, (i) Equation (5) additionally leverages the unused samples (*e.g.*, label-sparse samples) for unsupervised consistency regularization on data augmentations, which is known for improving generalization and feature learning even in supervised settings (Yang et al., 2022; Shen et al., 2022); (ii) meanwhile, it does not require prior knowledge of the sample subpopulations (*e.g.*, $\xi$ for $P_\xi$) which is essential for hard thresholding algorithms.

Equation (5) further facilitates the more flexible optimization process. As we will empirically show in Table 2, despite the close relation between Equation (5) and the hard thresholding algorithms (Remark 2), such updating strategies may be suboptimal for solving Equation (5).

---

**Algorithm 1** Adaptively Weighted Augmentation Consistency (*AdaWAC*)

---

1: **Input:** Training samples $\{(\mathbf{x}_i, \mathbf{y}_i)\}_{i \in [n]} \sim P_\xi^n$, augmentations $\{(A_{i,1}, A_{i,2})\}_{i \in [n]} \sim \mathcal{A}^{2n}$, maximum number of iterations $T \in \mathbb{N}$, learning rates $\eta_\theta, \eta_{\boldsymbol{\beta}} > 0$, pretrained initialization for the pixel-wise classifier $\theta_0 \in \mathcal{F}_{\theta^*}(\gamma)$.
2: Initialize the sample weights $\boldsymbol{\beta}_0 = \mathbf{1}/2 \in [0,1]^n$.
3: **for** $t = 1, \ldots, T$ **do**
4: $\quad$ Sample $i_t \sim [n]$ uniformly
5: $\quad$ $\mathbf{b} \leftarrow \left[ (\boldsymbol{\beta}_{t-1})_{[i_t]}, 1 - (\boldsymbol{\beta}_{t-1})_{[i_t]} \right]$
6: $\quad$ $\mathbf{b}_{[1]} \leftarrow \mathbf{b}_{[1]} \cdot \exp\left( \eta_{\boldsymbol{\beta}} \cdot \ell_{CE}\left( \theta_{t-1}; (\mathbf{x}_{i_t}, \mathbf{y}_{i_t}) \right) \right)$
7: $\quad$ $\mathbf{b}_{[2]} \leftarrow \mathbf{b}_{[2]} \cdot \exp\left( \eta_{\boldsymbol{\beta}} \cdot \ell_{AC}\left( \theta_{t-1}; \mathbf{x}_{i_t}, A_{i_t,1}, A_{i_t,2} \right) \right)$
8: $\quad$ $\boldsymbol{\beta}_t \leftarrow \boldsymbol{\beta}_{t-1}, (\boldsymbol{\beta}_t)_{[i_t]} \leftarrow \mathbf{b}_{[1]} / \|\mathbf{b}\|_1$
9: $\quad$ $\theta_t \leftarrow \theta_{t-1} - \eta_\theta \cdot \Big( (\boldsymbol{\beta}_t)_{[i_t]} \cdot \nabla_\theta \ell_{CE}\left( \theta_{t-1}; (\mathbf{x}_{i_t}, \mathbf{y}_{i_t}) \right)$
$\quad\quad\quad\quad\quad\quad\quad + \Big( 1 - (\boldsymbol{\beta}_t)_{[i_t]} \Big) \cdot \nabla_\theta \ell_{AC}\left( \theta_{t-1}; \mathbf{x}_{i_t}, A_{i_t,1}, A_{i_t,2} \right) \Big)$
10: **end for**

---

Inspired by the breakthrough made by Sagawa et al. (2020) in the distributionally-robust optimization (DRO) setting where gradient updating on weights is shown to enjoy better convergence guarantees than hard thresholding, in Algorithm 1, we introduce an adaptive weighting algorithm for solving Equation (5) based on online mirror descent. In contrast to the commonly used stochastic gradient descent (SGD), the flexibility of online mirror descent in choosing the associated norm space not only allows gradient updates on sample weights, but also grants distinct learning dynamics to sample weights $\boldsymbol{\beta}_t$ and model parameters $\theta_t$, which leads to the following convergence guarantee.

**Proposition 2** (Formally in Proposition 3, proof in Appendix B, assumptions instantiated in Example 1). *Assume that $\ell_{CE}(\theta; (\mathbf{x}, \mathbf{y}))$ and $\ell_{AC}(\theta; \mathbf{x}, A_1, A_2)$ are convex and continuous in $\theta$ for all $(\mathbf{x}, \mathbf{y}) \in \mathcal{X} \times [K]^d$ and $A_1, A_2 \sim \mathcal{A}^2$; $\mathcal{F}_{\theta^*}(\gamma) \subset \mathcal{F}_\theta$ is convex and compact. If there exist [4] (i) $C_{\theta,*} > 0$ such that $\frac{1}{n} \sum_{i=1}^n \left\| \nabla_\theta \widehat{L}_i^{WAC}(\theta, \boldsymbol{\beta}) \right\|_{\mathcal{F},*}^2 \leq C_{\theta,*}^2$ and (ii) $C_{\boldsymbol{\beta},*} > 0$ such that $\frac{1}{n} \sum_{i=1}^n \max\{ \ell_{CE}(\theta; (\mathbf{x}_i, \mathbf{y}_i)), \ell_{AC}(\theta; \mathbf{x}_i, A_{i,1}, A_{i,2}) \}^2 \leq C_{\boldsymbol{\beta},*}^2$ for all $\theta \in \mathcal{F}_{\theta^*}(\gamma)$, $\boldsymbol{\beta} \in [0,1]^n$, then with $\eta_\theta = \eta_{\boldsymbol{\beta}} = \frac{2}{\sqrt{5T(\gamma^2 C_{\theta,*}^2 + 2n C_{\boldsymbol{\beta},*}^2)}}$, Algorithm 1 provides*

$$
\mathbb{E}\left[ \max_{\boldsymbol{\beta} \in [0,1]^n} \widehat{L}^{WAC}(\overline{\theta}_T, \boldsymbol{\beta}) - \min_{\theta \in \mathcal{F}_{\theta^*}(\gamma)} \widehat{L}^{WAC}(\theta, \overline{\boldsymbol{\beta}}_T) \right] \leq 2\sqrt{5\left( \gamma^2 C_{\theta,*}^2 + 2n C_{\boldsymbol{\beta},*}^2 \right) \Big/ T}
$$

*where $\overline{\theta}_T = \frac{1}{T} \sum_{t=1}^T \theta_t$ and $\overline{\boldsymbol{\beta}}_T = \frac{1}{T} \sum_{t=1}^T \boldsymbol{\beta}_t$.*

---

[4]Following the convention, we use $*$ in subscription to denote the dual spaces. For instance, recalling the parameter space $\mathcal{F}_\theta$ characterized by the norm $\|\cdot\|_{\mathcal{F}}$ from Section 2.1, we use $\|\cdot\|_{\mathcal{F},*}$ to denote its dual norm; while $C_{\theta,*}, C_{\boldsymbol{\beta},*}$ upper bound the dual norms of the gradients with respect to $\theta$ and $\boldsymbol{\beta}$.

In addition to the convergence guarantee, Algorithm 1 also demonstrates superior performance over hard thresholding algorithms for segmentation problems in practice (Table 2). An intuitive explanation is that instead of filtering out all the label-sparse samples via hard thresholding, the adaptive weighting allows the model to learn from some sparse labels at the early epochs, while smoothly down-weighting $\ell_{CE}$ of these samples since learning sparse labels tends to be easier (Remark 1). With the learned model tested on a mixture of label-sparse and label-dense samples, learning sparse labels at the early stage is crucial for accurate segmentation.

## 5 EXPERIMENTS

In this section, we investigate the proposed *AdaWAC* algorithm (Algorithm 1) on different medical image segmentation tasks with different UNet-like architectures. We first demonstrate the performance improvements brought by *AdaWAC* in terms of sample efficiency and robustness to subpopulation shift (Table 1). Then, we verify the empirical advantage of *AdaWAC* compared to the closely related hard thresholding algorithms as discussed in Remark 2 (Table 2). Our ablation study (Table 4) further illustrates the indispensability of both sample reweighting and consistency regularization, the deliberate combination of which leads to the superior performance of *AdaWAC*[5].

**Experiment setup.** We conduct experiments on two volumetric medical image segmentation tasks: abdominal CT segmentation for Synapse multi-organ dataset (Synapse)[6] and cine-MRI segmentation for Automated cardiac diagnosis challenge dataset (ACDC)[7], with two UNet-like architectures: TransUNet (Chen et al., 2021) and UNet Ronneberger et al. (2015) (deferred to Appendix E.1). For the main experiments with TransUNet in Section 5, we follow the official implementation in (Chen et al., 2021) and use ERM+SGD as the baseline. We evaluate segmentations with two standard metrics—the average Dice-similarity coefficient (DSC) and average 95-percentile of Hausdorff distance (HD95). Dataset and implementation details are deferred to Appendix D. Given the sensitivity of medical image semantics to perturbations, our experiments only involve simple augmentations (*i.e.*, rotation and mirroring) adapted from (Chen et al., 2021).

### 5.1 SEGMENTATION PERFORMANCE OF *AdaWAC* WITH TRANSUNET

**Segmentation on Synapse.** Figure 2 visualizes the segmentation predictions on 6 Synapse test slices given by models trained via *AdaWAC* (ours) and via the baseline (ERM+SGD) with TransUNet (Chen et al., 2021). We observe that *AdaWAC* provides more accurate predictions on the segmentation boundaries and captures small organs better than the baseline.

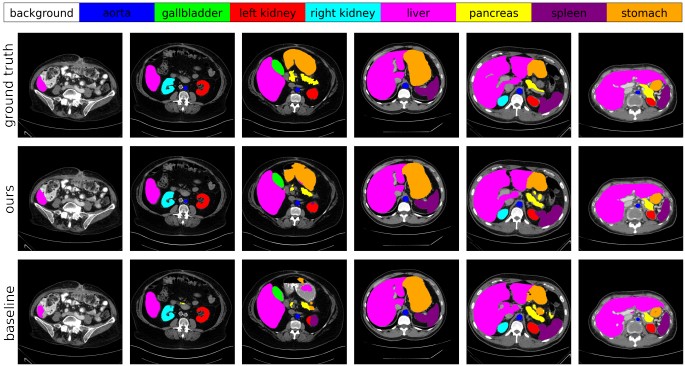

Figure 2: Visualization of segmentation predictions against the ground truth (in grayscale) on Synapse. Top to bottom: ground truth, ours (*AdaWAC*), baseline.

---

[5]We release our code anonymously at https://anonymous.4open.science/r/adawac-F5F8.

[6]https://www.synapse.org/#!Synapse:syn3193805/wiki/217789

[7]https://www.creatis.insa-lyon.fr/Challenge/acdc/

**Visualization of *AdaWAC*.** As shown in Figure 3, with $\ell_{CE}(\theta_t; (\mathbf{x}_i, \mathbf{y}_i))$ (Equation (2)) of label-sparse versus label-dense slices weakly separated in the early epochs, the model further learns to distinguish $\ell_{CE}(\theta_t; (\mathbf{x}_i, \mathbf{y}_i))$ of label-sparse/label-dense slices during training. By contrast, $\ell_{AC}(\theta_t; \mathbf{x}_i, A_{i,1}, A_{i,2})$ (Equation (3)) remains mixed for all the slices in the entire training process. As a result, the CE weights of label-sparse slices are much smaller than those of label-dense ones, pushing *AdaWAC* to learn more image representations but less pixel classification for slices with sparse labels and learn more pixel classification for slices with dense labels.

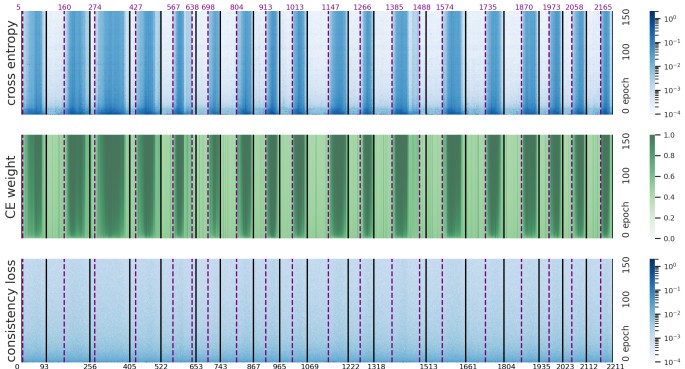

Figure 3: $\ell_{CE}(\theta_t; (\mathbf{x}_i, \mathbf{y}_i))$ (top), CE weights $\boldsymbol{\beta}_t$ (middle), and $\ell_{AC}(\theta_t; \mathbf{x}_i, A_{i,1}, A_{i,2})$ (bottom) of the entire Synapse training process. The x-axis indices slices 0–2211. The y-axis enumerates epochs 0–150. Individual volumes (cases) are partitioned by black lines; while the purple lines separate slices with/without non-background pixels.

**Sample efficiency and robustness.** We first demonstrate the *sample efficiency* of *AdaWAC* in comparison to the baseline (ERM+SGD) when training only on different subsets of the full Synapse training set ("**full**" in Table 1). Specifically, (i) **half-slice** contains slices with even indices only in each volume[8]; (ii) **half-vol** consists of 9 volumes uniformly sampled from the total 18 volumes in **full** where different volumes tend to have distinct $\xi$s (*i.e.*, ratios of label-dense samples); (iii) **half-sparse** takes the first half slices in each volume, most of which tend to be label-sparse (*i.e.*, $\xi$s are made to be small). As shown in Table 1, the model trained with *AdaWAC* on **half-slice** generalizes as well as a baseline model trained on **full**, if not better. Moreover, the **half-vol** and **half-sparse** experiments illustrate the *robustness* of *AdaWAC* to subpopulation shift. Furthermore, such sample efficiency and distributional robustness of *AdaWAC* extend to the more widely used UNet architecture. We defer the detailed results and discussions on UNet to Appendix E.1.

Table 1: *AdaWAC* with TransUNet trained on the full Synapse and its subsets.

| Training | Method | DSC ↑ | HD95 ↓ | Aorta | Gallbladder | Kidney (L) | Kidney (R) | Liver | Pancreas | Spleen | Stomach |
|---|---|---|---|---|---|---|---|---|---|---|---|
| full | baseline | $75.94 \pm 0.68$ | $32.91 \pm 8.80$ | 87.16 | 54.70 | 81.04 | 74.37 | 93.99 | 57.34 | 84.25 | 74.66 |
| | *AdaWAC* | $\mathbf{78.83 \pm 0.38}$ | $\mathbf{27.50 \pm 1.88}$ | 87.65 | 55.96 | 82.89 | 80.21 | 93.97 | 61.40 | 89.57 | 79.01 |
| half-slice | baseline | $74.62 \pm 0.78$ | $31.62 \pm 8.37$ | 86.14 | 44.23 | 79.09 | 78.46 | 93.50 | 55.78 | 84.54 | 75.24 |
| | *AdaWAC* | $\mathbf{77.37 \pm 0.40}$ | $\mathbf{29.56 \pm 1.09}$ | 86.89 | 55.96 | 82.15 | 78.63 | 94.34 | 57.36 | 86.60 | 77.05 |
| half-vol | baseline | $71.08 \pm 0.90$ | $46.83 \pm 2.91$ | 84.38 | 46.71 | 78.19 | 74.55 | 92.02 | 48.03 | 76.28 | 68.47 |
| | *AdaWAC* | $\mathbf{73.81 \pm 0.94}$ | $\mathbf{35.33 \pm 0.92}$ | 84.37 | 48.14 | 80.32 | 77.39 | 93.23 | 52.78 | 83.50 | 70.79 |
| half-sparse | baseline | $31.74 \pm 2.78$ | $69.72 \pm 1.37$ | 65.71 | 8.33 | 59.46 | 51.59 | 51.18 | 10.72 | 6.92 | 0.00 |
| | *AdaWAC* | $\mathbf{41.03 \pm 2.12}$ | $\mathbf{59.04 \pm 12.32}$ | 71.27 | 8.33 | 69.14 | 63.09 | 64.29 | 17.74 | 30.77 | 3.57 |

**Comparison with hard thresholding algorithms.** Table 2 illustrates the empirical advantage of *AdaWAC* over the hard thresholding algorithms, as suggested in Remark 2. In particular, we consider the following hard thresholding algorithms: (i) **trim-train** learns only from slices with at least one non-background pixel and trims the rest in each iteration on the fly; (ii) **trim-ratio** ranks the cross-entropy loss $\ell_{CE}(\theta_t; (\mathbf{x}_i, \mathbf{y}_i))$ in each iteration (mini-batch) and trims samples with the lowest cross-entropy losses at a fixed ratio – the ratio of all-background slices in the full training set

---

[8]Such sampling is equivalent to doubling the time interval between two consecutive scans or halving the scanning frequency in practice, resulting in the halving of sample size.

$(1 - \frac{1280}{2211} \approx 0.42)$; (iii) **ACR** further incorporates the augmentation consistency regularization directly via addition of $\ell_{AC}(\theta_t; \mathbf{x}_i, A_{i,1}, A_{i,2})$ without reweighting; (iv) **pseudo-*AdaWAC*** simulates the sample weights $\boldsymbol{\beta}$ at the saddle point and learns via $\ell_{CE}(\theta_t; (\mathbf{x}_i, \mathbf{y}_i))$ on slices with at least one non-background pixel while via $\ell_{AC}(\theta_t; \mathbf{x}_i, A_{i,1}, A_{i,2})$ otherwise. We notice that naive incorporation of **ACR** brings less observable boosts to the hard-thresholding methods. Therefore, the deliberate combination via reweighting in *AdaWAC* is essential for performance improvement.

Table 2: *AdaWAC* versus hard thresholding algorithms with TransUNet on Synapse.

| Method | baseline | trim-train | | trim-ratio | | pseudo-*AdaWAC* | *AdaWAC* |
|---|---|---|---|---|---|---|---|
| | | | +ACR | | +ACR | | |
| DSC ↑ | 76.28 | 76.01 | 75.66 | 74.26 | 77.98 | 78.28 | **79.12** |
| HD95 ↓ | 29.23 | 26.94 | 35.06 | 28.59 | 33.59 | 29.06 | **25.79** |

**Segmentation on ACDC.** Performance improvements granted by *AdaWAC* are also observed on the ACDC dataset (Table 3). We defer detailed visualization of ACDC segmentation to Appendix E.

Table 3: *AdaWAC* with TransUNet trained on ACDC.

| Method | DSC ↑ | HD95 ↓ | RV | Myo | LV |
|---|---|---|---|---|---|
| TransUNet | 89.36 | 3.02 | 88.36 | 83.84 | 95.87 |
| *AdaWAC* (ours) | **90.41** | **1.29** | **89.50** | **85.78** | **95.95** |

## 5.2 ABLATION STUDY

**On the influence of consistency regularization.** To illustrate the role of consistency regularization in *AdaWAC*, we consider the **reweight-only** scenario with $\lambda_{AC} = 0$ such that $\ell_{AC}(\theta_t; \mathbf{x}_i, A_{i,1}, A_{i,2}) \equiv 0$ and therefore $\mathbf{b}_{[2]}$ (Algorithm 1 line 7) remains intact. With zero consistency regularization in *AdaWAC*, reweighting alone brings little improvement (Table 4).

**On the influence of sample reweighting.** We then investigate the effect of sample reweighting under different reweighting learning rates $\eta_\beta$ (recall Algorithm 1): (i) **ACR-only** for $\eta_\beta = 0$ (equivalent to the naive addition of $\ell_{AC}(\theta_t; \mathbf{x}_i, A_{i,1}, A_{i,2})$), (ii) ***AdaWAC*-0.01** for $\eta_\beta = 0.01$, and (iii) ***AdaWAC*-1.0** for $\eta_\beta = 1.0$. As Table 4 implies, when removing reweighting from *AdaWAC*, augmentation consistency regularization alone improves DSC slightly from 76.28 (baseline) to 77.89 (ACR-only), whereas *AdaWAC* boosts DSC to 79.12 (*AdaWAC*-1.0) with a proper choice of $\eta_\beta$.

Table 4: Ablation study of *AdaWAC* with TransUNet trained on Synapse.

| Method | DSC ↑ | HD95 ↓ | Aorta | Gallbladder | Kidney (L) | Kidney (R) | Liver | Pancreas | Spleen | Stomach |
|---|---|---|---|---|---|---|---|---|---|---|
| baseline | 76.28 | 29.23 | 87.46 | 55.21 | 82.06 | 77.76 | 94.10 | 54.06 | 85.07 | 74.54 |
| reweight-only | 76.68 | 29.24 | 86.15 | 53.98 | 82.96 | 80.28 | 93.42 | 55.86 | 85.29 | 75.49 |
| ACR-only | 77.89 | 31.65 | 87.96 | 54.34 | 81.79 | 80.21 | 94.52 | 60.41 | 88.07 | 75.83 |
| *AdaWAC*-0.01 | 77.94 | 27.81 | 87.58 | 52.75 | 82.29 | 80.22 | 94.90 | 55.92 | 91.63 | 78.23 |
| *AdaWAC*-1.0 | **79.12** | **25.79** | 87.23 | 54.94 | 84.58 | 81.69 | 94.62 | 58.29 | 90.63 | 81.01 |

## 6 DISCUSSION

In this paper, we exploit the non-uniformity in labels commonly observed in volumetric medical image segmentation via *AdaWAC*—a deliberate combination of adaptive weighting and augmentation consistency regularization. By casting the separation between sparse and dense segmentation labels as a subpopulation shift in the label distribution, we leverage the unsupervised consistency regularization on encoder layer outputs (of UNet architectures) as a natural reference to distinguish label-sparse and label-dense samples via comparisons against the supervised average cross-entropy losses. We formulate such comparisons as a weighted augmentation consistency (WAC) regularization problem and propose an adaptive weighting scheme—*AdaWAC*—for iterative and smooth separation of samples from different subpopulations with a convergence guarantee. Our experiments demonstrate empirical effectiveness of *AdaWAC* not only in improving segmentation performance and sample efficiency but also in enhancing robustness to the subpopulation shift in labels.

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
