# OpenReview forum: "AdaWAC: Adaptively Weighted Augmentation Consistency Regularization for Volumetric Medical Image Segmentation"
_ICLR.cc/2023/Conference — Submitted to ICLR 2023_

### Official Review · Reviewer_VPhg · 2022-10-24

**Confidence:** 3
**Correctness:** 3
**Technical Novelty And Significance:** 3
**Empirical Novelty And Significance:** 3
**Recommendation:** 5

**Clarity, Quality, Novelty And Reproducibility:**

This paper is well-written and easy to follow. The idea is interesting and new. Experiments are presented with details.

**Strength And Weaknesses:**

Strength
+ An interesting solution to consistency regularization
+ Comparison with hard thresholding algorithms
+ Good study on robustness

Weakness
+ no comparison with the soft thresholding algorithm


**Summary Of The Paper:**

This paper proposes a new method, namely adaptively weighted augmentation consistency (AdaWAC), by combining adaptive weighting and augmentation consistency regularization. It models the separation between sparse and dense labels as a subpopulation shift in the label distribution. It is demonstrated in the experiments that the segmentation performance has been improved in terms of DSC and HD95.

**Summary Of The Review:**

The paper presents an interesting idea and good experimental results. However, the evaluation needs improvement.

---

> ### Author Response · Authors · 2022-11-13
> **Response to Reviewer VPhg**
>
> Thanks a lot for appreciating our ideas and experiments. On the concern about evaluation raised in the review:
>
> [__C1. Comparison with “soft thresholding algorithms”__] First, to avoid potential misunderstanding of the question, we would appreciate further clarification from the reviewer on what “soft thresholding algorithms” refer to.
>
> According to our understanding, the “soft thresholding” counterparts of the hard thresholding algorithms in Table 2 will be adaptive sample reweighting algorithms like [1] that down-weight samples instead of trimming the sample set strictly. In this case, the method “reweight-only” in our ablation study (Table 4) is exactly doing “soft thresholding”. As the results in Table 4 suggested, “soft thresholding”/reweighting alone brings little improvement compared to the baseline. But when coupling “soft thresholding”/reweighting with consistency regularization via AdaWAC, larger improvements can be achieved.
>
> [1] Sagawa, Shiori, Pang Wei Koh, Tatsunori B. Hashimoto, and Percy Liang. "Distributionally robust neural networks for group shifts: On the importance of regularization for worst-case generalization." arXiv preprint arXiv:1911.08731 (2019).

---

### Official Review · Reviewer_QgJJ · 2022-11-02

**Confidence:** 5
**Clarity, Quality, Novelty And Reproducibility:** The manuscript is of moderate quality…
**Correctness:** 3
**Technical Novelty And Significance:** 2
**Empirical Novelty And Significance:** 2
**Recommendation:** 3

**Details Of Ethics Concerns:**

N/A.

**Strength And Weaknesses:**

<Strength>

•	The paper is well-organized and well-written.

•	The experimental results supports the claims made in the manuscripts.

•	The submitted content is related to the application of deep neural networks in 3D medical image segmentation, which is highly relevant to the ICLR audience.

<Weakness>

•	The novelty of the proposed algorithm is limited. The WAC regularization is similar to the existing works like MixMatch or FixMatch.

•	The effectiveness and practical values of the proposed algorithm is unconvincing. Only a single 2D backbone neural network is
studied in the experiments. And the results of the proposed algorithm are not compared to other state-of-the-art segmentation algorithms on the same datasets or public leaderboards.

•	It is lack of comparison of other semi-supervised learning algorithms in medical image segmentation.

•	Discussion or ablation studies are not sufficient (see below).

<Comments>

1.	How do the (nearly) failure cases look like, especially the cases that the proposed algorithm cannot fix?

2.	Does the batch size matter for the performance of Algorithm 1?

3.	Would using a different random seed in model training give similar results?

4.	It would be necessary to conduct ablation studies on the effects of different backbone networks.

5.	How are the losses computed in the Table 1 for the baselines with partial annotations? Are the losses calculated only on the annotated
areas?

6.	In Table 2, ACR is able to help improve Dice score for “trim-ratio”. Why does it hurt the Hausdorff distance a lot? Where is the improvement of Dice score from?

7.	It would be better to evaluate the algorithm via the public testing leaderboard of the Synapse dataset.

8.	The experiments are based on a 2D segmentation networks for 3D image segmentation tasks. It is necessary to conduct the experiments with 3D networks as well.


**Summary Of The Paper:**

The manuscript proposed an adaptive data weighting algorithm for 3D medical image segmentation. The proposed algorithm dynamically adjusts weights between supervised loss and self-supervised consistency loss during model training. The weights are adjusted by the values of loss terms on-the-fly. And the self-supervised loss is based on different augmentations on the same images, and introduced as weighted augmentation consistency (WAC) regularization. Moreover, the experimental results validate the proposed algorithm on two public 3D medical image segmentation datasets. Both fully- and partially-labeled scenarios are studied in the experiments.

**Summary Of The Review:**

My concerns about the manuscript are about its novelties and practical impacts. The novelties are limited because the WAC regularization is similar to the existing works like MixMatch or FixMatch and no comparison of such works were conducted. Moreover, only a single 2D backbone neural network is studied in the experiments. The results of the proposed algorithm are not compared to other state-of-the-art segmentation algorithms on the same datasets or public leaderboards.

---

> ### Author Response · Authors · 2022-11-13
> **Response to Reviewer QgJJ**
>
> Many thanks for the detailed comments and questions. First, we would like to kindly point out some misunderstandings in the review concerning our setting and novelty, as explained in the general response G1.
>
> On the concerns and questions raised in the review:
>
> [__B1. Novelty__: The WAC regularization is similar to the existing works like MixMatch or FixMatch.] Please refer to the general response G1.
>
> [__B2. Experiments on 3D networks__: It is necessary to conduct the experiments with 3D networks as well.] We want to kindly emphasize that __AdaWAC is designed to improve the classical combination of ERM+SGD for segmenting volumetric medical images as 2D slices__ instead of 3D volumes. The motivation and key assumption of our method is the __subpopulation shift in the segmentation labels of 2D slices in each 3D volume__ (i.e., the separation between sparse and dense labels), which is not satisfied by the 3D segmentation problem.
>
> [__B3. Comparison with SOTA__] Please refer to the general response G2.
>
> [__B4. Comparison with semi-supervised learning algorithms__] Despite the usage of augmentation consistency regularization (which is well-known for its semi-supervised learning ability), in this work, we are considering the __fully supervised__ setting. We believe that the supervised setting is valuable especially for medical image applications because, in contrast to data for general vision tasks, medical data are usually limited in both raw images and labels [1,2] (e.g., due to privacy concerns and regulations). Since our method does not involve additional unlabeled data, it does not make sense to make comparisons with semi-supervised learning algorithms.
>
> [__B5. Visualization of (nearly) failure cases__] We will add a visualization section for the near failure cases to the appendix in our revision.
>
> [__B6. Batch size__: Does the batch size matter for the performance of Algorithm 1?] From the practical perspective, yes. As for most online optimization algorithms, the larger batch size gives lower variance, which usually leads to better performance in practice (a standard argument). In the experiments, we chose the same batch size as the previous work [5] (which is 24) for fair comparisons.
>
> However, from the algorithmic perspective, Algorithm 1 holds regardless of the batch size (i.e., samples in each batch are processed independently). In this case, we followed the common practice (e.g., in the related works [3,4]) and presented the algorithm in its online version.
>
> [__B7. Random seeds__: Would using a different random seed in model training give similar results?] Yes, we have tested the main experiments in Table 1 (full) with three arbitrary random seeds and observed very similar performance gaps between the baseline and AdaWAC (refer to the first table in the general response G3). In the paper, we follow the common practice in [5] and present results given by the same random seed used in the TransUNet experiments. We agree with the reviewer that including the averages and standard deviations of multiple trials will be helpful. Due to the time limit for the rebuttal, we may not be able to make such updates for all our experiments. But we will make sure to include these experiments in a future revision.
>
> [__B8. Experiments on different backbone networks__] We appreciate the constructive question from the reviewer. In the revision, we will add the results of the main experiments on UNet, which coincided with our empirical observations on TransUNet (refer to the second table in the general response G3).

---

> > ### Author Response · Authors · 2022-11-13
> > **Additional response to Reviewer QgJJ**
> >
> > [__B9__. How are the losses computed in Table 1 for the baselines with partial annotations? Are the losses calculated only on the annotated areas?] Thanks for raising this confusion. We would like to clarify that “half” in Table 1 refers to a subset of slices (instead of annotations) in the original dataset, where the number of slices/volumes is half the number of original slices/volumes, but not annotations within slices are halved. Hence, the losses are calculated on the subset of slices that we sampled with full annotations.
> >
> > [__B10. Difference between DSC and HD95__: In Table 2, ACR is able to help improve DSC for “trim-ratio”. Why does it hurt the Hausdorff distance a lot? Where is the improvement of Dice score from?] It is worth noticing that __DSC and Hausdorff distance (HD95) are measuring different aspects of the segmentation predictions__. As discussed in Appendix C, DSC is measuring the pixel-wise alignment of the prediction and the ground truth in each class, without taking the relative position of pixels into consideration. By contrast, Hausdorff distance is measuring the alignment of segmentation boundaries. To see their difference, consider a simple yet common scenario in medical image segmentation where the prediction mislabels a very small set of background pixels as an organ (e.g., pancreas), and the mislabeled pixels are far away from the ground truth pancreas pixels. In this scenario, since the mislabeled set is small, DSC will not be affected much; but since the mislabeled pixels are far away from the ground truth, HD95 will be compromised a lot due to the misclassifications. Due to such sensitivity of Hausdorff distance to outliers, HD95 is commonly used as a supplementary metric aside from DSC.
> >
> > [1] Asgari Taghanaki, Saeid, Kumar Abhishek, Joseph Paul Cohen, Julien Cohen-Adad, and Ghassan Hamarneh. "Deep semantic segmentation of natural and medical images: a review." Artificial Intelligence Review 54, no. 1 (2021): 137-178.
> >
> > [2] Karimi, Davood, Haoran Dou, Simon K. Warfield, and Ali Gholipour. "Deep learning with noisy labels: Exploring techniques and remedies in medical image analysis." Medical Image Analysis 65 (2020): 101759.
> >
> > [3] Sagawa, Shiori, Pang Wei Koh, Tatsunori B. Hashimoto, and Percy Liang. "Distributionally robust neural networks for group shifts: On the importance of regularization for worst-case generalization." arXiv preprint arXiv:1911.08731 (2019).
> >
> > [4] Wu, Xiaoxia, Yuege Xie, Simon Shaolei Du, and Rachel Ward. "Adaloss: A computationally-efficient and provably convergent adaptive gradient method." In Proceedings of the AAAI Conference on Artificial Intelligence, vol. 36, no. 8, pp. 8691-8699. 2022.
> >
> > [5] Chen, Jieneng, Yongyi Lu, Qihang Yu, Xiangde Luo, Ehsan Adeli, Yan Wang, Le Lu, Alan L. Yuille, and Yuyin Zhou. "Transunet: Transformers make strong encoders for medical image segmentation." arXiv preprint arXiv:2102.04306 (2021).

---

### Official Review · Reviewer_epcY · 2022-11-03

**Confidence:** 3
**Correctness:** 3
**Technical Novelty And Significance:** 3
**Empirical Novelty And Significance:** 2
**Recommendation:** 5

**Clarity, Quality, Novelty And Reproducibility:**

The paper organization and the paper writing are poor, which makes the paper difficult to understand.

**Strength And Weaknesses:**

Ads:
This paper proposes the AdaWAC as an adaptive weighting algorithm
that assigns label-dense samples to supervised cross-entropy loss and label-sparse
samples to unsupervised consistency regularization.
This paper also proposes a WAC regularization that uses the consistency of encoder layer outputs as a natural reference to incentivize separation between samples with sparse and dense labels.
Experiments are conducted to validate the results showing the effectiveness of the proposed methods.

Dis:
The paper organization and the paper writing are poor, which makes the paper difficult to understand.
In general, there is a lack of explanation of the methods used in the study.
Furthermore, an explanation of why the authors did these experiments should be provided.
In experimental table 1, the AdaWAC proposed in this paper is still worse than the baseline results in some situations. The authors are advised to provide some explanations and justifications.
The results of comparing current SOTA methods used in medical image segmentation should be presented. Now the article only shows the results of the baseline and the proposed AdaWAC.


**Summary Of The Paper:**

This paper proposes the AdaWAC as an adaptive weighting algorithm
that assigns label-dense samples to supervised cross-entropy loss and label-sparse
samples to unsupervised consistency regularization.
This paper also proposes a WAC regularization that uses the consistency of encoder layer outputs as a natural reference to incentivize separation between samples with sparse and dense label.

**Summary Of The Review:**

This paper proposed AdaWAC as an adaptive weighting algorithm
that assigns label-dense samples to supervised cross-entropy loss and label-sparse
samples to unsupervised consistency regularization.
However, this paper is difficult to understand.

---

> ### Author Response · Authors · 2022-11-13
> **Response to Reviewer epcY**
>
> Thank you for appreciating our ideas and methods. To clarify the organization of our work, we hope that the following __roadmap__ will be helpful. Overall, we present AdaWAC in the respective order of the main contributions listed at the end of the introduction.
>
> - In Section 2, we formalize the discrepancy between the sparse and dense labels within each volume as a subpopulation shift and introduce the problem setup (including the augmentation consistency regularization).
> - In Section 3, we introduce the weighted augmentation consistency (WAC) problem in Eq(5) and show that WAC provides separation between samples with sparse and dense labels at the saddle point in Prop 1.
> - In Section 4, we present the AdaWAC algorithm (Algo 1) for solving the WAC problem via adaptive sample reweighting and provide the convergence guarantee in Prop 2.
> - In Section 5, we empirically demonstrate that AdaWAC not only enhances segmentation performance and sample efficiency but also improves distributional robustness.
>
> We will add this roadmap to the introduction in the revision.
>
> On the detailed concerns and questions raised in the review:
>
> [__A1. Averaged v.s. organ-wise DSC__: In Table 1, AdaWAC is worse than baseline on some organs.] As explained in Appendix D, the Synapse dataset has 9 classes, including the background and 8 organs. In Table 1 (as well as Table 4-7),  following the common practice [1,2], DSCs are evaluated for individual organs separately, and the goal is to achieve a higher __averaged DSC across all organs__ (in the column ‘DSC’) instead of improving DSCs for all organs. Of course, it would have been ideal if DSCs can be improved for all classes, which is challenging in practice (cf. [1,2] where [1] improved DSC on 5 out of 8 organs and [2] achieved improvements on 4 out of 8 organs). In our experiments (e.g., the updated results in the first table of general response G3), the DSCs of AdaWAC are at least as good as (within standard deviations), but generally better than, those of the baseline for all organs.
>
> [__A2. Comparison with SOTA__: The results of comparing current SOTA methods used in medical image segmentation should be presented.] Please refer to the general response G2.
>
> [1] Chen, Jieneng, Yongyi Lu, Qihang Yu, Xiangde Luo, Ehsan Adeli, Yan Wang, Le Lu, Alan L. Yuille, and Yuyin Zhou. "Transunet: Transformers make strong encoders for medical image segmentation." arXiv preprint arXiv:2102.04306 (2021).
>
> [2] Cao, Hu, Yueyue Wang, Joy Chen, Dongsheng Jiang, Xiaopeng Zhang, Qi Tian, and Manning Wang. "Swin-unet: Unet-like pure transformer for medical image segmentation." arXiv preprint arXiv:2105.05537 (2021).

---

### Author Response · Authors · 2022-11-13
**General response**

We thank all the reviewers for their valuable comments and suggestions. Here, we address some critical concerns raised by the reviewers that we believe are caused by misunderstanding. We kindly hope that this will help the reviewers have a better understanding of our work.

[__G1. Novelty: difference compared to MixMatch/FixMatch__] We would like to highlight that both the setting and the algorithm described in our work are completely different from those of MixMatch/FixMatch, despite the common idea of consistency regularization on data augmentations (which is popular and inspiring numerous works, as discussed in the related works (Section 1.1)).
- In terms of setting, we are considering the __supervised setting__ whereas MixMatch and FixMatch are designed for the semi-supervised setting.
- A key challenge of incorporating augmentation consistency regularization in segmentation tasks is that, in contrast to classification labels, __the segmentation labels are not invariant to common augmentations like rotation and mirroring__. Without this assumption on the invariance of labels to augmentations, consistency regularization methods for classification tasks like MixMatch/FixMatch cannot be naively adapted to the segmentation setting. For medical image segmentation, in the related works, we review some strategies for overcoming this challenge by enforcing consistency in the segmentation output space (e.g., [1]). By contrast, __our algorithm leverages the insensitivity of sparse labels to augmentations and encourages consistent encodings on label-sparse samples__.
- In contrast to Imagenet data, medical images are inherently unsuitable for strong augmentations like the MixUp used in MixMatch and Cutout/RandAugment used in FixMatch. These __strong augmentations are essential for MixMatch/FixMatch but tend to destroy the semantic information in medical images__. Following the standard practice in the medical image regime (e.g., [1,2]), we use weak augmentations like random rotation and mirroring only.
- The key novelty of our work is not only on consistency regularization but also (more importantly) on the __careful combination of sample reweighting and consistency regularization__, which is fundamentally different from MixMatch/FixMatch.

[__G2. Comparison with SOTA__]
It is worth noticing that __AdaWAC is an optimization algorithm designed to improve the classical combination of ERM+SGD and is widely applicable to different backbone architectures__. Therefore, the goal of our experiments is to demonstrate that AdaWAC is capable of improving over ERM+SGD for commonly used backbone architectures like the UNet [3] (UNet experiments will be included in the revision) and TransUNet [2] (which is one of the SOTA backbone architectures), instead of chasing the SOTA performance.

Meanwhile, the current SOTA algorithms on the Synapse leaderboard are either semi-supervised algorithms that leverage additional unlabeled data or 3D segmentation networks that do not fit in our problem setup. Therefore, we chose well-tested 2D segmentation architectures like the UNet and TransUNet as the backbone of our experiments.

In addition to comparisons on the full Synapse dataset, via experiments on partial datasets in Table 1, we also demonstrate the sample efficiency and distributional robustness of AdaWAC, which are essential advantages in practice beyond the measurement of the leaderboard.

[1] Li, Xiaomeng, Lequan Yu, Hao Chen, Chi-Wing Fu, Lei Xing, and Pheng-Ann Heng. "Transformation-consistent self-ensembling model for semisupervised medical image segmentation." IEEE Transactions on Neural Networks and Learning Systems 32, no. 2 (2020): 523-534.

[2] Chen, Jieneng, Yongyi Lu, Qihang Yu, Xiangde Luo, Ehsan Adeli, Yan Wang, Le Lu, Alan L. Yuille, and Yuyin Zhou. "Transunet: Transformers make strong encoders for medical image segmentation." arXiv preprint arXiv:2102.04306 (2021).

[3] Ronneberger, Olaf, Philipp Fischer, and Thomas Brox. "U-net: Convolutional networks for biomedical image segmentation." In International Conference on Medical image computing and computer-assisted intervention, pp. 234-241. Springer, Cham, 2015.

---

> ### Author Response · Authors · 2022-11-13
> **General response: additional experiments**
>
> [__G3. Additional experiments__] Here, we highlight some key updates on experiments that we will add in the revision.
>
> 1. Updates for main experiments on __TransUNet__ (in Table 1) with averages over 3 different random seeds.
>
> | Method | DSC | HD95 | Aorta | Gallbladder | Kidney (L) | Kidney (R) | Liver | Pancreas | Spleen | Stomach |
> | :---: | :---: | :---: | :---: | :---: | :---: | :---: | :---: | :---: | :---: | :---: |
> | baseline | 75.94 $\pm$ 0.68 | 32.91 $\pm$ 8.80 | 87.16 | 54.70 | 81.04 | 74.37 | 93.99 | 57.34 | 84.25 | 74.66 |
> | AdaWAC | __78.83 $\pm$ 0.38__ | __27.50 $\pm$ 1.88__ | 87.65 | 55.96 | 82.89 | 80.21 | 93.97 | 61.40 | 89.57 | 79.01 |
>
> 2. New main experiments on __UNet__ with averages over 3 different random seeds.
>
> | Method | DSC | HD95 | Aorta | Gallbladder | Kidney (L) | Kidney (R) | Liver | Pancreas | Spleen | Stomach |
> | :---: | :---: | :---: | :---: | :---: | :---: | :---: | :---: | :---: | :---: | :---: |
> | baseline | 74.04 $\pm$ 1.52 | 36.65 $\pm$ 0.33 | 84.93 | 55.59 | 77.59 | 70.92 | 92.21 | 55.01 | 82.87 | 73.21 |
> | AdaWAC | __76.71 $\pm$ 0.62__ | __30.67 $\pm$ 2.85__ | 85.53 | 59.26 | 79.44 | 74.81 | 94.21 | 56.95 | 86.87 | 80.16 |

---

### Author Response · Authors · 2022-11-17
**Summary of updates**

We thank all the reviewers again for their constructive and inspiring comments, which have helped us identify problems in our work and make improvements accordingly. We would also like to apologize for the delay in revision due to the irreducible time for running new experiments. As an overview of the revision, we summarize the major updates as follows.
1. In Appendix E.1, we add __experiments with UNet__ whose results coincide with our observations with TransUNet. These additional experiments demonstrate the __flexibility of the proposed AdaWAC framework in the choice of backbone architectures__.
2. In Table 2, we add the __comparison of AdaWAC with a new type of hard thresholding algorithm which we call ‘pseudo-AdaWAC’__. Specifically, pseudo-AdaWAC simulates the sample weights at the saddle point and learns via the supervised cross-entropy loss on slices with at least one non-background pixel while via the unsupervised augmentation consistency regularization otherwise. Compared to the other hard thresholding methods (trim-train and trim-ratio), pseudo-AdaWAC is more closely related to AdaWAC. To be precise, pseudo-AdaWAC leverages the exact idea motivating the WAC formulation but learns without adaptive reweighting. The new results in Table 2 illustrate that pseudo-AdaWAC achieves better performance than trim-train and trim-ratio, but not as good as AdaWAC. This suggests both the utility of the WAC formulation and the necessity of the adaptive sample reweighting in AdaWAC.
3. In Appendix E.3, we add __visualization of segmentation on Synapse with severe distributional shift__ (i.e., when trained on the half-sparse subset), with associated discussions.
4. In Table 1, we update the __main experiments__ by replacing the previous results based on the original random seed used in the TransUNet experiments with the __averages (and standard deviations) over 3 arbitrary random seeds__. It is worth mentioning that while experimenting with different random seeds, we found that the model collapse of the baseline (ERM+SGD) when training on the half-sparse subset of Synapse can be avoided with careful hyper-parameter (learning rate) tuning, albeit the worse performance than AdaWAC. For fair comparisons, both updates in Appendix E.3 and in Table 1 present baseline results without collapses and the corresponding AdaWAC results in the same settings.

---

### Decision · Program_Chairs · 2023-01-20

**Decision:**

Reject

**Justification For Why Not Higher Score:**

See the summary

**Justification For Why Not Lower Score:**

N/A.

**Metareview: Summary, Strengths And Weaknesses:**

This paper presents a new regularization term (ADAPTIVELY WEIGHTED AUGMENTATION CONSISTENCY REGULARIZATION) for volumetric segmentation in medical images. Based on my reading, the paper can be understood, but the presentation can be improved for easier understanding and highlight the main contributions. A major weakness is the validation as the authors only use Unet and Trans-Unet as baselines, lacking of comparison with state-of-the-art approaches. The reviewers got some point in the concern that the authors shall compare with some state-of-the-art algorithms. Although comparison with Unet and TransUnet show some improvement, it is important to see if such improvement is consistent with latest methods.

Another concern is the experimental justification. Although the title suggests that the approach is for volumetric segmentation. The actual experimental results are for segmenting volumetric medical images as 2D slices. This is quite misleading. Some experiments based on 3D network shall be conducted.

**Summary Of Ac-Reviewer Meeting:**

N/A